# Titanium and Silicon Dioxide-Coated Fabrics for Management and Tuning of Infrared Radiation

**DOI:** 10.3390/s22103918

**Published:** 2022-05-22

**Authors:** Ismail Yuce, Suat Canoglu, Sevhan Muge Yukseloglu, Roberto Li Voti, Gianmario Cesarini, Concita Sibilia, Maria Cristina Larciprete

**Affiliations:** 1Textile, Clothing, Shoe and Leather Department, Technical Science Vocational School, Trakya University, Edirne 22030, Turkey; ismailyuce@trakya.edu.tr; 2Department of Textile Engineering, Faculty of Technology, Marmara University, Istanbul 34722, Turkey; scanoglu@marmara.edu.tr (S.C.); myukseloglu@marmara.edu.tr (S.M.Y.); 3Dipartimento di Scienze di Base ed Applicate per l’Ingegneria, Sapienza Università di Roma, Via Antonio Scarpa 16, 00161 Rome, Italy; gianmario.cesarini@uniroma1.it (G.C.); concita.sibilia@uniroma1.it (C.S.); mariacristina.larciprete@uniroma1.it (M.C.L.)

**Keywords:** smart textiles, infrared emissivity, infrared manipulation, infrared thermography, FTIR

## Abstract

Far infrared radiation (FIR) is emitted by every body at a given temperature, including the human body. FIR ranging between 4–14 μm is considered useful for cell growth, and the human body emits a maximum of infrared (IR) radiation at the wavelength of approximately 9.3 µm. In the present study, fabrics based on five different raw textiles having the same yarn count as well as the same weaving patterns were designed and created. Some of them were subjected to a coating process. The fabrics to be tested were as follows: coated with TiO_2_ nanoparticles, coated with SiO_2_ nanoparticles, coated fabric that does not contain bioceramic nanoparticle (BNFC), and non-coated fabrics (NCF). The structural characterization of the resulting samples was performed using scanning electron microscopy (SEM), abrasion tests, and air permeability. Following the structural characterization, the infrared emissivity properties were investigated using infrared thermography as well as attenuated total reflectance Fourier-transform infrared spectroscopy in the 8–14 IR range. According to the experimental findings, the fabrics coated with TiO_2_ and SiO_2_ displayed increased infrared emissivity values compared to the uncoated ones. In addition, it was observed that the use of bioceramic powders had no effect on air permeability and abrasion properties.

## 1. Introduction

Recent years have witnessed growing interest in far-infrared (FIR) textiles, an emerging category of functional textiles for health and comfort applications. Typically, the functionalization of FIR textiles is reached by using bioceramic additives in the form of micro- or nano-particles. Oxide-based materials such as alumina, silica or tantalia can emit infrared radiation which has the effects of improving blood circulation and stimulating the lymphatic system.

The present study focuses on the investigation of the structural and functional properties of woven fabrics composed of different yarns and coated with nanoceramic powders.

The term infrared radiation (IR) includes electromagnetic waves in the wide wavelength range ranging between visible light and terahertz frequencies. Several studies have shown that the FIR radiation in the wavelength range of 4–14 µm produces beneficial effects for human health [1,2,3] and plants [4,5]. The human body itself, according to Planck’s black-body emission law, emits IR radiation with a peak wavelength of about 9.3 µm [6].

In the last years, the use of IR radiation for therapeutic purposes has witnessed an increasing interest. Fabrics with IR-emitting features have been developed and widely employed for sportswear, medical textiles, home textiles, and clothes [7]. Several additives can be used for the preparation of smart textiles featuring IR emissive properties such as metallic yarns [8,9] carbon nanotubes [10,11] or phase-change microcapsules [12], to name a few. In particular, bioceramic powders are extensively used for the production of IR-emitting textiles [13,14,15,16,17], since these micron-sized powders can be added into textiles during fiber spinning or by coating [14,18,19,20]. 

Ceramic nanoparticles can be integrated into textile products in two general ways. The first method is by integrating nanoparticles into the fibers during the spinning process. In order to do this, first the nanoparticles are brought into masterbatch form and mixed with fiber spinning raw materials in the extruder and then added to the fibers [14,19,20]. In the second method, after the fabric is obtained, ceramic nanoparticles can be added to the textile products by such methods as spraying, coating, laminating and impregnating, and covering and dipping [14,18]. In this way, fabrics obtained from natural fibers (cotton, wool, etc.) can be given FIR features. Such a procedure can be also applied to high emissivity carbon nanofiber-laminated fabrics, so as to produce a further increase in the emissivity properties in the FIR, as shown in recent papers [21].

The effects of bioceramic textiles on human health have been investigated in the literature on athletes, women, and patients with heart failure. In a study, the benefits of FIR tights were dermatologically tested for the reduction of cellulite and local adiposity defects. The study was conducted on 42 women aged 20–60 with cellulite and local adiposity problems [20]. A total of 22 women wore IR-featured tights, and the others tested regular tights with the same model and shape. According to the results, it was observed that the users of FIR tights experienced a reduction of cellulite and local adiposity defects. It was also observed that the IR-featured tights improved the skin elasticity, skin smoothness, and skin compactness [20].

A simple explanation of such beneficial effects on health is as follows: the bioceramics increase thermal insulation by reflecting the IR radiation emitted from the body back to the body surface. The IR radiation emitted by the human body is transferred to the ceramic particles that act as “*absorbers*” and then radiate the IR back to the body surface [22]. Several researchers have investigated how bioceramic additives affect the thermal properties of textiles, and have shown that bioceramics absorb IR rays emitted from the body and retain heat. For these reasons, the temperature difference between the body and the clothing is reduced, preserving the body’s heat [23]. Silicon dioxide (SiO_2_) and titanium dioxide (TiO_2_) are ideal nanoparticles for the emission of electromagnetic radiation in the range 8-14 μm, which is coincident with the emission range of the human body [24].

In the present study, we investigated cotton, wool, viscose, acrylic, and cotton/polyester woven fabrics covered with different ceramic powder coatings for tuning and managing the IR emissivity. Samples’ structural characterization included scanning electron micrography, abrasion resistance, air permeability as well as ATR-FTIR analysis. Concerning the functional characterization, the effect of different bioceramic powders on IR emissivity was experimentally investigated using IR thermography for emissivity measurements. Experimental findings show the noticeable effect of the content of ceramic powders on tuning IR emissivity.

## 2. Materials and Methods

### Sample Preparation

Bioceramic fabric samples were created using five different yarn types such as 100% cotton, 100% acrylic, 100% viscose, 100% wool, and 50-50% cotton/polyester which have the same yarn count and the same weaving construction. The count values of the yarns (supplied by Yunsa and Akren Yarn) were provided as 30 tex and then made into double yarn. In order to get woven fabrics with the same structural properties (thickness, fabric density, and weaving construction), all yarn counts were chosen to be the same, i.e., 1 × 1 plain weave cloth. The weaving process was carried out using a weaving machine (Uğur Tekstil corporation) by preparing the warps side by side on the warp loom and picking the weft yarns, respectively. Concerning fibers’ properties, the measured weaving density and weight values of the investigated fabrics are reported in Table 1. Following fabric samples preparation, the hydrophilization process was carried out at 95 °C for 45 min and at 50 °C for 30 min using a Haspel machine (Roaches) for cotton and cotton/polyester raw yarns. Hydrophilization of the wool fabric was carried out at 45 °C for 4 h. The wool hydrophilization was performed using following parameters: H_2_O_2_ (%35): 30 mL/L, Sodium silicate: 7 g/L, pH: 8.5–9.

After the hydrophilization process, the investigated fabrics were coated with different bioceramic nanoparticles. Specifically, two coatings including titanium dioxide (TiO_2_) and silicon dioxide (SiO_2_) nanoparticles were prepared respectively, and a coating without bioceramic nanoparticles (BNFC) was made. For each type of fabric, along with coated fabrics, uncoated fabrics (NCF) were also characterized and tested. The dimensions of the bioceramic nanoparticles, the degree of purity, and the viscosity of the prepared coating resin are given in Table 2.

The coating resin was prepared by means of a mixer (Janke & Kunkel RW20) to amalgamate the powder and resin. The coating was composed of several chemicals supplied by Rudolf Duraner: Acrylate (Rudolf Duraner AC111), cross-linker (RUCO-COAT FX 8011), thickener (RUCO-COAT TH 5020), dispersing agent (Rudolf Duraner AD 719). The details on the composition of coating chemicals and bioceramic powder are reported in Table 2.

The coating process was carried out using the Ataç GK40 RKL device (at Marmara University), and the coated fabrics were kept at 165 °C for 2 min for the drying and fixing procedure.

## 3. Structural Characterization

### 3.1. SEM Images

As a first structural characterization, the presence of bioceramic nanoparticles in the coated fabrics was observed and confirmed using scanning electron microscopy (SEM). SEM images of the coated fabrics were recorded using the JEOL JSM-IT100 equipment within Marmara University Faculty of Technology Textile Engineering. The effective transfer of bioceramic powders onto fabrics was confirmed by SEM images. In Figure 1, it is possible to observe SEM images of some of the uncoated fabrics (100% Cotton, 100% Wool, and 100% Viscose), and the related SiO_2_ nanoparticles coated fabrics. In Figure 2, analogous SEM images are for some of the uncoated fabrics (Cotton/PES and 100% Acrylic) and the related TiO_2_ nanoparticles coated fabrics. From the SEM images investigation, it was observed that particles were not homogeneously distributed all over the fabrics, and particles size was different in all coatings. Furthermore, from the same SEM images, it was possible to retrieve bioceramic particle sizes using a suitable computer program (ImageJ). The sizes of bioceramic additives observed from image investigations were about 1–2 μm, much larger than the particle size reported in Table 2, showing the strong effects of particle aggregation. However such aggregation did not affect the resulting IR properties of the investigated coated fabrics, as will be clear in the next section.

### 3.2. Abrasion Test

The abrasion test was carried out using the Martindale abrasion test device at Marmara University. An abrasion resistance test is carried out to determine the permanence and waterproofing properties of the coatings, which directly affects their durability. The test was applied, using the mass-loss method, to bioceramic nanoparticle free coatings (BNFC) as well as to coatings containing TiO_2_ and coatings containing SiO_2_. Abrasion resistance refers to the pulling resistance of fabric in the process of repeated friction. According to the test standard (TSE EN ISO 12947-1), three pieces of each fabric were prepared using a circular sampling apparatus and then loaded onto the lower plates of the Martindale abrasion tester. The abrading disk was then rubbed against the sample in an oscillating circle. Tests were carried out setting a pressure value of 12 kPa (795 ± 7 g) that is typically employed for technical use fabrics. The fabrics were cut in ∅ 38 mm circular shapes and 3 pieces were taken from each sample. Both 5000 and 10,000 rubs cycles of abrasion were performed on the samples. Each sample was weighed on a precision scale before and after testing, respectively, and % weight loss values were accurately evaluated after abrasion cycles.

The abrasion test results on different samples show that all fabrics had low mass loss values after 5000 or 10,000 rub cycles. Specifically, the % mass variation at 5000 rubs cycles was found to be less than 1% for all samples except those of wool fabrics. Only woolen fabrics displayed values higher than 2% which increased to about 4% after 10,000 rub cycles (Figure 3). The highest abrasion loss in wool fabrics can be ascribed to their high moisture content. A second reason is that the abrasive fabric (bottom fabric) used in the Martindale device is a type of 100% wool fabric. Consequently, the epidermis layer of wool fibers causes them to be worn more in friction.

The coating type also affected the abrasion values as it was observed that the abrasion resistance of the fabrics decreased as the moisture content increased [25]. In general, the percent of weight loss of the samples coated with SiO_2_ reached higher values with respect to TiO_2_ and BNFC. At the same time, the percent of the weight loss of samples coated with TiO_2_ was higher than BNFC.

The observed differences can be explained by the increase in the viscosity value of the coating paste when nanoparticles are added as the abrasion weight loss increases with increasing viscosity values [26]. Indeed, in a previous study of cotton woven fabrics coated with SiO_2_ and Al_2_O_3_ particles, it was determined that the type of solution used in the coating affected the resulting abrasion resistance, whereas the presence of particles did not play a significant role [27]. After the abrasion test, the water repellency test was applied to the fabric to evaluate the loss of SiO_2_ and TiO_2_ on the fabric. SiO_2_ and TiO_2_ are particles that can impart water repellency to fabrics [28]. Total particle loss was measured by modeling by taking SEM images of coated fabrics before and after abrasion.

### 3.3. Air Permeability Test

Air permeability takes into account the amount of air passing through the fibers, yarns, and fabric structure at unit pressure, per unit area and unit time [29]. The air permeability test was carried out with the Prowhite Airtest II K008 device at Mimar Sinan University Vocational School Clothing Production Technology. Tests were carried out according to ISO 9237 standard, at 100 Pa pressure, 20 cm^2^ sample area, and 27 °C room temperature.

The air permeability values of the samples are displayed in the graph shown in Figure 4. From the obtained experimental results, it was observed that the fabric type with the highest air permeability values in all samples were measured in woolen fabrics. The second highest air permeability samples were from viscose fabrics. The lowest air permeability values were observed in acrylic fabrics (Figure 4). The high air permeability of woolen fabrics can be related to the physical structure of the fibers. We can attribute the high air permeability of viscose fabrics to its low weight value (Table 1) [29]. 

In general, after coating, air permeability is reduced because the coating chemical fills the spaces between the fibers and yarns [30], therefore, the air permeability values of the fabrics that do not contain coating chemicals should display purple bars value higher than those containing them.

In agreement with these considerations, viscose and acrylic have higher purple bars than the others. The cotton and cotton/PES fabrics were bleached, thus, their air permeability values before and after coating were found to be very similar. At the same time, wool was pre-treated (washed), and the air permeability values were higher than the others due to its physical structure.

Concerning the different particles, the air permeability values of the samples coated with titanium dioxide were found to be higher compared to those coated with silicon dioxide. This is a consequence of the high viscosity of the silicon dioxide coating resin as well as the larger particle size with respect to the titanium dioxide particles.

## 4. Functional Characterization

### 4.1. Infrared Emissivity Measurements

The infrared emission of the textiles was characterized in the IR range (8–14 μm) by observing their temperature evolution under a heating regime with a focal plane array (FPA) IR camera [31]. A standard test method for measuring and compensating emissivity using IR imaging radiometers as well as a reference surface of known higher emissivity was applied to the set of investigated fabrics. The experimental results were interpreted by means of Plank’s theory of black-body radiation, and thus, the IR emissivity at different applied temperatures was retrieved.

Measurements were carried out with an IR camera (COX CX320 Thermal Camera) at Sapienza University laboratories. According to ASTM E1933-99a standards for IR emissivity measurements [32], a graphite paint (Bonderite L-GP 386 Acheson) characterized by a well-known IR emissivity value (black body, ε ≈ 0.98) was employed as a reference sample [12].

The scheme of the experimental apparatus used for the measurement of the IR emissivity is shown in Figure 5. Here, the textile sample was placed in direct contact with a thermoregulator plate equipped with both a heating and Peltier modulus, acting as heating and cooling source, respectively, allowing to set the temperature from 10 °C up to a maximum heating temperature of 150 °C. At the same time, two contact-thermometers (T-type thermocouples) were put in direct contact with both the sample and plate surface, respectively, in order to accurately monitor their actual temperature during the measurement run. The measurements on the different fabrics were carried out by recording thermographic images at three different temperatures of the heating plate (45 °C, 60 °C, and 80 °C) that were always higher than the room temperature (24 °C). Such a choice was made due to the following:(i)The IR signal from the heated sample comes from two main sources: the sample emittance and the environment. If the sample temperature is higher than the room temperature, the second becomes negligible, and consequently, the noise in the emissivity measurement decreases.(ii)The difference of at least 20 °C between sample and room temperatures guarantees the accurate emissivity measurement of textiles.(iii)The standard textile emissivity is expected to be constant with temperature in the range from 20 °C to 80 °C, and in conditions of low humidity. Therefore, it is convenient to estimate the sample emissivity by measuring the IR signals from heated samples at relatively high temperatures with respect to room temperature.(iv)Only metal-based textiles [8,9], or textiles containing thermochromic or phase change materials [33], or a regular structured metasurface could exhibit a temperature-dependent emissivity [34].

At the end of each temperature run, the recorded thermographic images are processed by means of a dedicated software, which enables the possibility to select the image pixeled areas corresponding to either the sample or the reference surface. By analyzing the apparent temperature read by the IR camera along with the real temperature read by thermocouples, the average emissivity value can be eventually estimated.

As an example, Figure 6 shows the comparison among three acrylic fabric samples (coated with SiO_2_, TiO_2_, and uncoated), which were placed close to each other so as to highlight the differences. Figure 6a shows the apparent temperature of the three textiles where the real temperature measured by the thermocouple was 45 °C. From the color map, one can see that the uncoated acrylic sample exhibits a dark orange color corresponding to an apparent temperature that is lower than that of the the coated textiles. After data processing, the apparent temperature map in Figure 6a can be turned into the emissivity map in Figure 6b, showing that acrylic fabrics coated with SiO_2_ and TiO_2_ exhibit a rather homogenous surface with an average emissivity of 0.85, and 0.86 respectively, to be compared with the lower value of 0.76 corresponding to the uncoated acrylic fabric (see also Table 3).

In order to verify that, in these textiles, emissivity is almost temperature independent, several measurements were performed approximatively at 45 °C, 60 °C, and 80 °C. Figure 7 shows the emissivity measurements vs. temperature for the same investigated samples in Figure 6. The emissivity value is averaged over a 1 cm^2^ area and is measured three times for each temperature (see symbols in Figure 7). As can be observed in Figure 7, the overall trend of the emissivity vs. temperature is almost constant in the temperature range up to 80 °C, as expected. Moreover, it is confirmed that the average emissivity values vary from 0.76, corresponding to bare acrylic fabric (Figure 7c), to the higher emissivity values of 0.85 and 0.86 for the SiO_2_ (Figure 7a) and TiO_2_-coated(Figure 7b) acrylic fabric.

The averaged emissivity values measured following the described procedure for all investigated fabric samples are summarized in Table 3.

The obtained experimental results show that the emissivity values of the coated fabric samples increased compared to the uncoated fabrics. Furthermore, for cotton and acrylic fabrics, a slight increase was also observed once bare coating (BNFC samples) was replaced with the bioceramic nanoparticles. According to the obtained results, coating seems to be an effective tool for modifying and tuning the resulting IR emissivity of investigated textiles, and all investigated coatings appear to be suitable for IR emissivity modulation applications. Note that the emissivity values found for BNFC are usually higher than the ones for NCF due to the fact that the additional contribution of the binder in the coating chemical increases the emissivity value [35,36]. It is also worth noting that the anisotropy of the IR emissivity or thermal parameters was not revealed for these samples [37,38].

### 4.2. ATR-FTIR Analysis

The effect of bioceramic nanoparticles on the IR spectral properties of the resulting fabrics was investigated using attenuated total reflectance Fourier-transform IR spectroscopy (ATR-FTIR).

First of all, the ATR-FTIR spectra of different fabrics shown in Figure 8 highlight how the 5 types of “*non-coated*” fabrics display dissimilar IR spectral features. On one side, cotton-based and viscose fabrics show a strong absorption peak at about 10 μm, which makes them absorbent and therefore highly emissive in the LWIR (emission values 0.80–0.84, in Table 3). On the other hand, acrylic and wool fabrics display an almost constant absorption band over the whole investigated IR range, although it is not very intense and without evident strong peaks. As a consequence, the measured emissivity (ε) values in the LWIR is lower for 100% wool (ε = 0.75) and 100% acrylic textiles (ε = 0.76).

In Figure 9a, the FTIR spectra of TiO_2_-coated fabrics are shown. Some vibration peaks in the wavelength range of 4–14 μm, which are not visible in the uncoated samples, are attributed to the Ti-O bond (Figure 9a) [39,40]. As an example, in Figure 9b, the details of the ATR-FTIR graphs of TiO_2_-coated (solid lines) and BNFC (dotted lines) wool fabrics are reported to highlight the effect of TiO_2_ onto FTIRs spectra. In the measured spectra, the peaks in the wavelength region of 2.88 μm (3740 cm^−1^) are attributed to the hydroxyl group (OH) of TiO_2_ [41], whereas the peak occurring at about 7.3 μm (1371–1377 cm^−1^) in the TiO_2_-coated samples is ascribed to the Ti-H bond [42]. Finally, the peak at about 11.8 μm (780–830 cm^−1^) reveals the Ti-O bond [43]. On the other hand, the coating also introduced the strong peak occurring at a wavelength of 5.78 μm (1728 cm^−1^), which is due to the vibration of residual water molecules (H-O-H) [44,45].

The FTIR spectra of all fabrics coated with SiO_2_ are displayed in Figure 10a. Figure 10b shows details of the ATR-FTIR curves corresponding to SiO_2_-coated (solid lines) and BNFC (dotted lines) acrylic fabrics. From these graphs, it can be observed that the coated fabric shows the peaks occurring at a wavelength of 9 μm and 12.7 μm which correspond to the Si-O-Si bond (1100 cm^−1^ and 786 cm^−1^) [44,45]. As for TiO_2_-coated fabrics, we can attribute the sharp peak located at 1726 cm^−1^ to water molecules (H-O-H).

The presence of bioceramics also affects the resulting spectral features since binders also added some characteristic peaks between 850 and 1600 cm^−1^, which is clearly visible in almost all the spectra of the 10 samples. This leads to increased absorption in the LWIR range and the consequent enhancement observed in the emissive properties (Table 3 and Table 4).

In order to find a correlation between the emissivity values and absorbance spectra in the LWIR range, the emissivity enhancement Δ*ε*, due to the use of bioceramic nanoparticles, is correlated to the equivalent enhancement of the absorbance spectrum Δ*A(**λ)*. The correlation can be found when the Planck’s black body spectrum equation is applied to the absorption/emission properties of the bioceramic fabric versus the reference (bare) fabric as follows:(1)$$\Delta \varepsilon_{IR}=\frac{\int_{\lambda_{\min}}^{\lambda_{\max}}\Delta A\left(\lambda \right)\cdot \frac{2hc^{2}}{\lambda^{5}}\frac{1}{\exp\left(\frac{hc}{\lambda kT}\right)-1}d\lambda }{\int_{\lambda_{\min}}^{\lambda_{\max}}\frac{2hc^{2}}{\lambda^{5}}\frac{1}{\exp\left(\frac{hc}{\lambda kT}\right)-1}d\lambda }$$
where *h* is the Planck constant, *k* the Boltzmann constant, *c* the speed of light, *λ*_min_ = 8 μm, and *λ*_max_ = 14 μm are the boundaries of the operational wavelength window of the IR camera. By inserting the differential absorbance spectrum Δ*A*(*λ*) measured with ATR-FTIR in Equation (1), it is therefore possible to evaluate the theoretical values of Δ*ε* that an IR camera should ideally record. From the comparison between the differential contributions of emissivity Δ*ε* calculated from the ATR-FTIR absorbance spectra, reported in Table 4, and those measured with an IR thermal imaging camera (Table 3), it can be seen that for many additives, there is a general agreement within a tolerance of about ±0.03, which is in reasonable agreement with the combined measurement errors for the two different measurement techniques. Any major discrepancies may be explained by taking into account that samples do not have a high homogeneity of deposition, thus, results may therefore differ if measurements are performed at different positions of the fabric samples.

Local correlations among the content of bioceramic nanopartcicles, the IR emissivity, and the acoustic and mechanical properties of the textiles can be found by using other spatially and time resolved experimental techniques [46,47,48].

## 5. Conclusions

In an attempt to maximize the IR radiation absorbed by our bodies, yarns and fabrics with peculiar emissive properties are investigated and developed as well as sold for commercial applications. In the present paper, we outline our experimental investigation of 5 woven fabrics consisting of different raw materials/yarns. Two different bioceramic coatings, based on TiO_2_ and SiO_2_ nanoparticles, were added to the fabric surface, and both their structural and emissive properties, before and after coating, were experimentally characterized. Using SEM micrographs, it was possible to observe that the bioceramic nanoparticles were efficiently grafted onto the fabrics; however, aggregation effects were also found. Furthermore, the mechanical stability of the fabrics was tested using the abrasion test. According to the experimental findings, weight losses were highest in woolen coatings, whereas it was rather low for all other coatings, showing a good mechanical adhesion of the coating to the fabric. The wool fabric also showed the highest air permeability, whereas the lowest air permeability values were observed for acrylic fabrics. After the structural characterization, the IR emissive properties were investigated using both thermal micrographs and ATR-FITR interferometry. The results were consistent, showing that the bioceramic powders added to the fabric efficiently modulates the IR emissivity in the investigated wavelength range (8–14 microns), as bioceramic additives increase the emissivity values in the IR range. The results obtained open up potential applications to improve body health and thermal comfort, but also in agriculture, to accelerate plant growth, as recently shown in References [4,5]. It is worth noting that the procedure for adding bioceramic nanoparticles can be accomplished at relatively low cost, with no significant increase in the total production costs of such smart fabrics compared to standard commercially available ones.

## Figures and Tables

**Figure 1 sensors-22-03918-f001:**
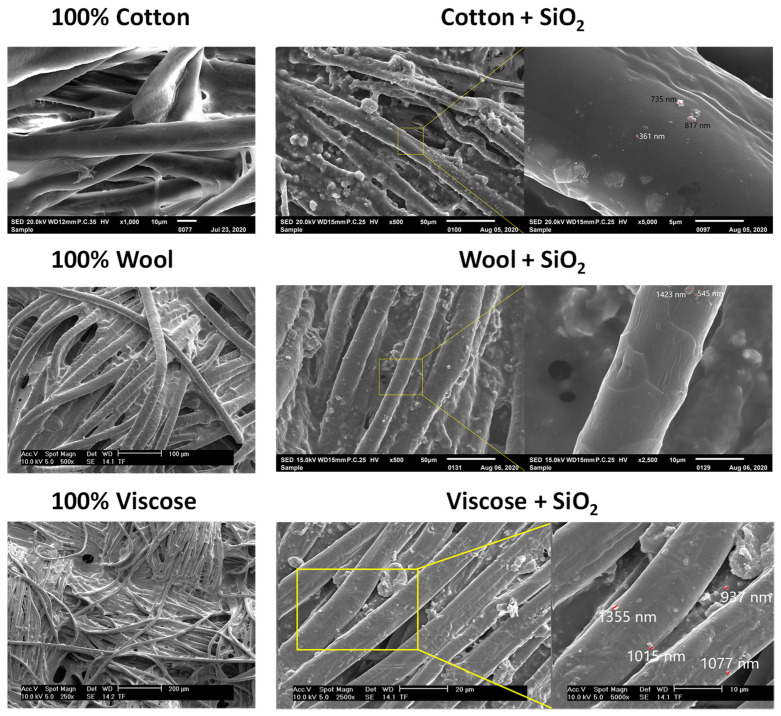
SEM images of uncoated fabrics and related SiO_2_-coated fabrics.

**Figure 2 sensors-22-03918-f002:**
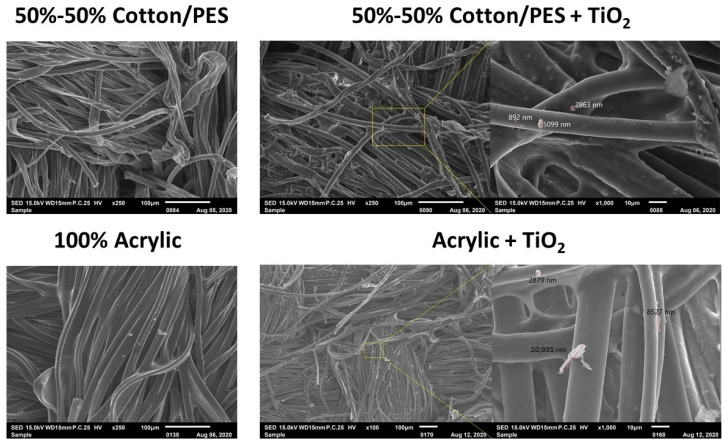
SEM images of uncoated fabrics and related TiO_2_-coated fabrics.

**Figure 3 sensors-22-03918-f003:**
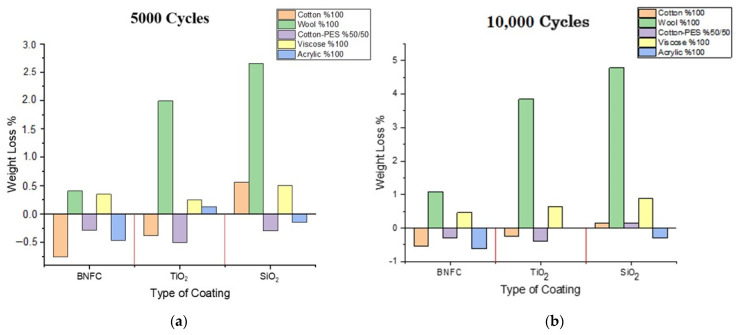
Percent weight loss: (**a**) after 5000 cycles; (**b**) after 10,000 cycles.

**Figure 4 sensors-22-03918-f004:**
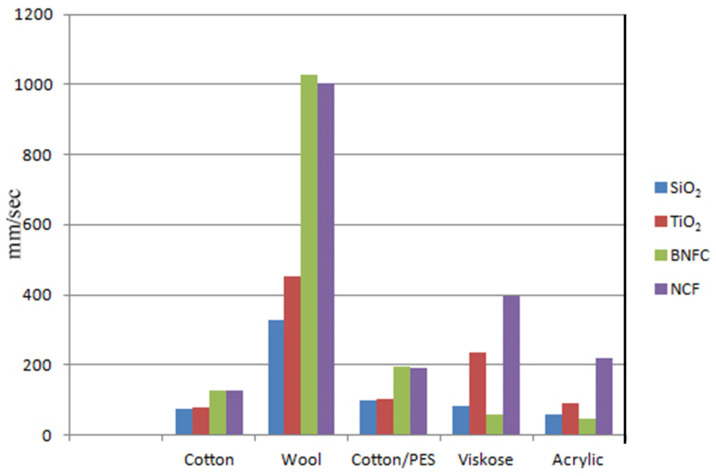
Air permeability graph of samples.

**Figure 5 sensors-22-03918-f005:**
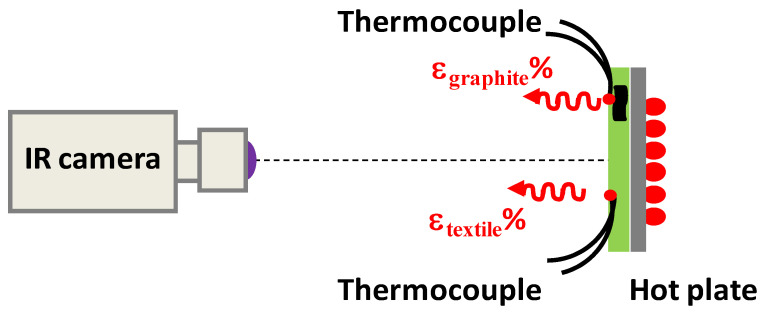
Sketch of the experimental setup employed for the characterization of the temperature-dependent bioceramic fabrics emissivity.

**Figure 6 sensors-22-03918-f006:**
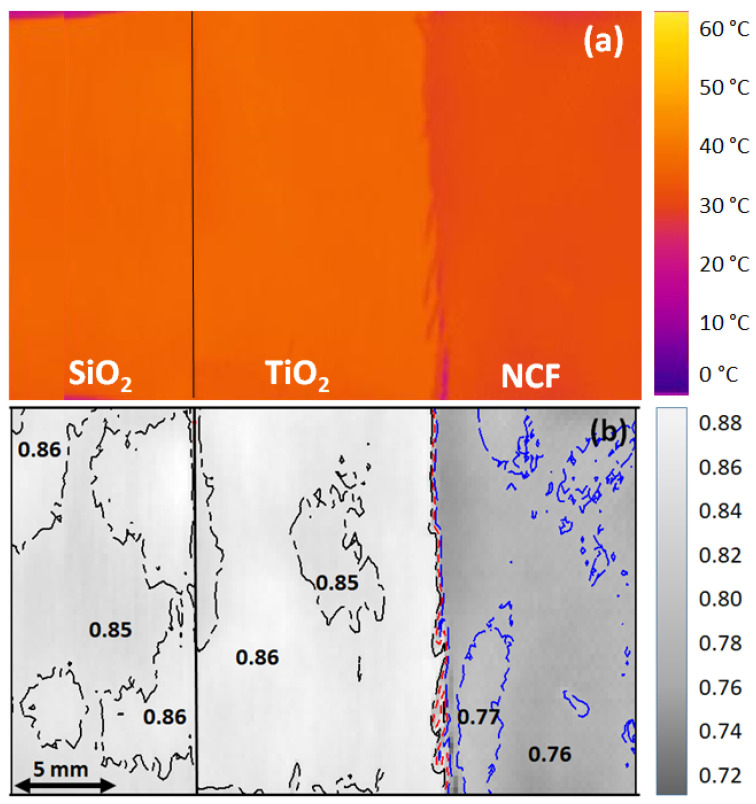
Compnteredarison among three acrylic fabric samples placed close to each other: coated with SiO_2_ and TiO_2_ bioceramic nanoparticles, and uncoated. The thermal imager captures all samples together: (**a**) apparent temperature. The temperature of each textile is at 45 °C, as measured by the thermocouple; (**b**) textile emissivity map, as calculated from the thermal image in Figure 5a.

**Figure 7 sensors-22-03918-f007:**
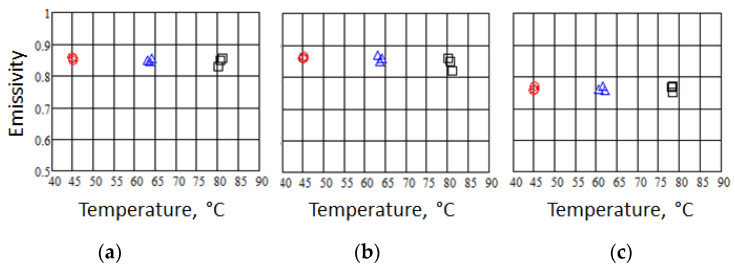
Large scale emissivity measurements vs. temperature of acrylic fabric samples: (**a**) acrylic coated with SiO_2_ nanoparticles; (**b**) acrylic coated with TiO_2_ nanoparticles; and (**c**) acrylic uncoated. The emissivity is averaged over 1cm^2^ area and is measured three times for each temperature. The textiles were heated at the temperature: (o) 45 °C; (Δ) 60 °C; and (□) 80 °C.

**Figure 8 sensors-22-03918-f008:**
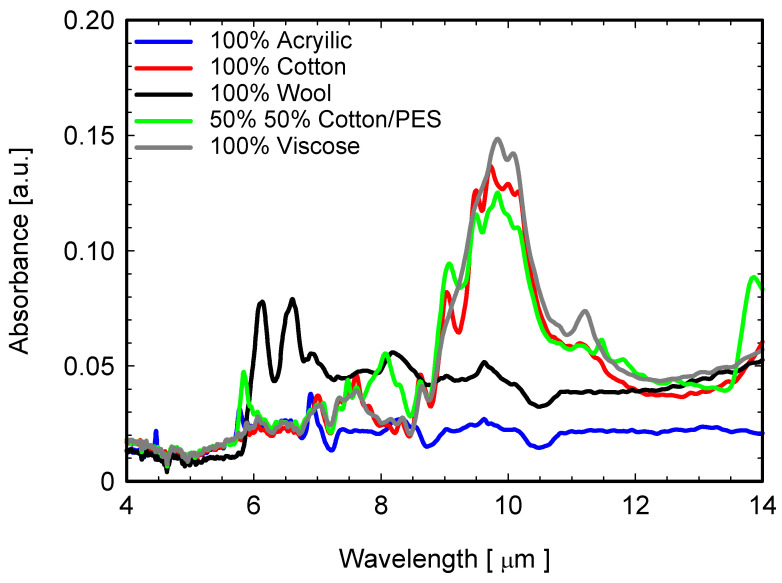
ATR-FTIR spectra of investigated bare fabrics. Inset: details of the ATR-FTIR graphs of uncoated fabrics in the range 4–14 μm.

**Figure 9 sensors-22-03918-f009:**
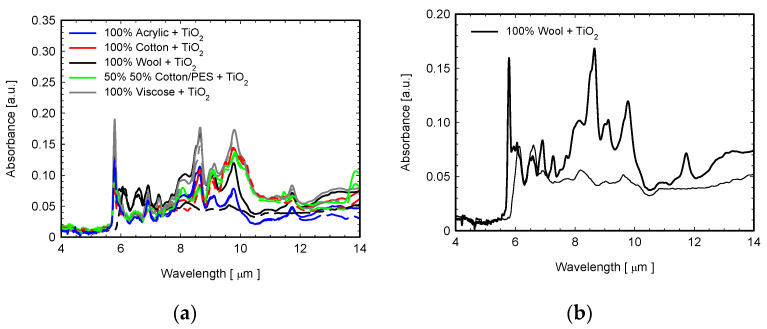
(**a**) ATR-FTIR spectra of investigated fabrics coated with TiO_2_ (solid lines) and uncoated (dashed lines) in the range 4–14 μm; (**b**) detailed comparison between coated (solid line) and uncoated (dotted line) wool fabrics.

**Figure 10 sensors-22-03918-f010:**
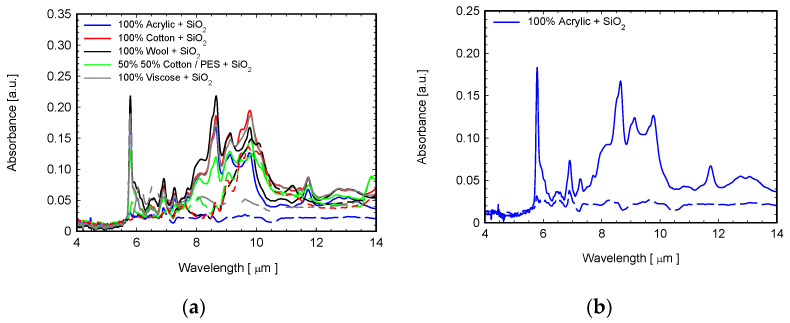
(**a**) ATR-FTIR spectra of investigated fabrics coated with SiO_2_ (solid lines) and uncoated (dashed lines) in the range 4–14 μm; (**b**) detailed comparison between coated (solid line) and uncoated (dotted line) acrylic fabrics.

**Table 1 sensors-22-03918-t001:** Weaving density and weight density values of the yarns employed for samples realization.

Fabric Type	Weft Density (Piece/cm)	Warp Density (Piece/cm)	Surface Mass Density (g/m^2^)
100% Cotton	13	14	227
100% Wool	14	15	194
100% Acrylic	14	16	176
100% Viscose	14	15	170
50-50% Cotton/PES	14	15	212

**Table 2 sensors-22-03918-t002:** Chemical additives and bioceramic powders features.

Chemical Used	Percent (%)	Bioceramic Particle Size and Purity Grade	Viscosity (Pa.s)
Acrylate (Rudolf Duraner AC111)	20	-	-
Water	75.5	-	-
Cross-linker (RUCO-COAT FX 8011)	1.5	-	-
Dispersing agent (Rudolf Duraner AD 719)	0.5	-	-
Thickener (RUCO-COAT TH 5020)	1.5	-	-
SiO_2_	1	55–75 nm, 98.5%	9.3
TiO_2_	1	17 nm, 99.9%	6.2
BNFC	-	-	2.2

**Table 3 sensors-22-03918-t003:** Measured emissivity (ε) values of samples.

	NCF	BNFC	TiO_2_	SiO_2_
Cotton (100%)	0.80	0.81	0.84	0.85
Wool (100%)	0.75	0.84	0.83	0.81
Cotton-PES (50-50%)	0.84	0.85	0.83	0.83
Viscose (100%)	0.84	0.86	0.86	0.88
Acrylic (100%)	0.76	0.76	0.86	0.85

**Table 4 sensors-22-03918-t004:** Differential contributions of emissivity Δ*ε_IR_* calculated from the ATR-FTIR absorbance spectra using Equation (1).

	NCF	BNFC	TiO_2_	SiO_2_
Cotton (100%)	Reference	0.03	0.05	0.07
Wool (100%)	Reference	0.01	0.06	0.12
Cotton-PES (50-50%)	Reference	0.03	0.04	0.03
Viscose (100%)	Reference	0.06	0.06	0.08
Acrylic (100%)	Reference	0.06	0.06	0.11

## Data Availability

Research data are not shared.

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
