# Peer review of "Titanium and Silicon Dioxide-Coated Fabrics for Management and Tuning of Infrared Radiation"

_sensors, 2022, doi:10.3390/s22103918_

Round 1
Reviewer 1 Report
In this paper, the author prepared infrared radiation coating fabrics using TO2, SiO2 or bioceramic nanoparticles to improve the body health and thermal comfort. However, the relationship between structure and performance still needs to be improved. There are some suggestions:
- It is difficultly to judge the innovation in this work, without the comparison of the resent research progress in the infrared radiation of nanoparticles and coating fabrics.
- The infrared radiation of TO2, SiO2 nanoparticles compared of that of carbon material ( such as graphene),What are the advances?
- The more information about the prepared infrared radiation coating fabrics should be added, such as the scheme of prepared.
- What about the result of the different amount of nanoparticle used in the coating?
- How about the washing fastness and the performance after washing or abrasion?
- Why the temperature was heated to 50-90 oC? it is far above the human body temperature.
- The relationship between structure and performance still needs to be improved, the mechanism should be given. Why the fabrics coated with TiO2 and SiO2 display increased infrared emissivity values? why the use of bioceramics powders had no effect on air permeability and abrasion properties?
- Figure 1, 8 and 9 is not clear enough to distinguish the information and curves. Other figures have similar problems, it should be improved.
- What about production costs in comparison with commercially available ones?
- What about the potential applications of the produced infrared radiation coating fabrics?
Author Response
Please find enclosed the reply to the reviewer #1 in attachment

Reviewer 2 Report
1. Please, correct typos in the chemical formula. (e.g. subscript)
2. The SEM images in Figure 1 are not clear. please, replace them with high-resolution ones.
3. IR stands for infrared.
4. Please, delete the empty area in Figure 3.
5. There is no spacing between °C and value.
6. Figure 5 shows the thermographic images of the only cotton sample. It will be better if all other fabric results are compared. And, again, the images in Figure 5 are not clear. please, replace them with high-resolution ones.
7. In Table 3, BNFC samples of Wool and cotton samples show the highest emissivity. Explain why.
8. The ART-FTIR analysis in Figure 8 and 9 are not clearly explained. Please, mark the specific peak or peak ranges you want to emphasize.
9. Please, add the laundry test results for the coated samples.
Author Response
please find enclosed the reply to reviewer #2 in attachmnet

Reviewer 3 Report
The authors are requested to clarify the following points in the revised manuscript.
- Due to the small size of nanoparticles, it is difficult to be metabolized by the human body once absorbed by the human body. Therefore, there seems to be a certain safety hazard in the coating of nanomaterials on fabric on fabrics that are worn all year round. Please give some comments about this. Besides, titanium dioxide nanoparticles and silicon dioxide nanoparticles do not seem to be considered bioceramic powders.
- Please check Figure 1 carefully, some SEM images do not have a scale bar. How many grams of titanium dioxide nanoparticles or silicon dioxide nanoparticles are loaded on a piece of fabric per unit area?
- In Figure 2, what proportion of titanium dioxide nanoparticles or silicon dioxide nanoparticles are left after the abrasion test, compared with that before the abrasion test? The ratio to the total mass (fabric + nanoparticles) does not seem to be meaningful.
- In Figure 3, for Viscose and Acrylic, why is the BNFC much lower than the NCF?
- In Figure 5, the temperature selection of the hot plate is 50, 70, and 90 degrees Celsius, which is much higher than the temperature range of the human body, why? What is the performance at normal human body temperature?
- What do the red, blue and black dots in Figure 6 mean? Why is the emissivity so close at different temperatures?
- Please change the abscissa of Figure 7, 8 and 9 to the unit of wavelength, which makes the readers more accessible.
Author Response
please find enclosed the reply to the reviewer #3 in attachment

Round 2
Reviewer 1 Report
The manuscript is accepted for publication in its revised form.
Author Response
We would like to thank you Reviewer #1 for providing constructive criticism and opportunity to improve the quality and the strength of our manuscript.
We are happy to read now tha our manuscript is now judged suitable for publication in the present revised form
Reviewer 3 Report
The authors took into consideration my suggestions and appropriately revised their manuscript, which I think now could be accepted for publication in Sensors after a minor revision.
1, There is still an SEM image without a scale bar in Figure 2.
2, The mass loading of TiO2/SiO2 particles on the fabric is quite important. Please provide at least one idea of how this can be measured.
3, Please check the text in the manuscript carefully, for example, in Figure 9a and 10a, there are some overlapping words.
Author Response
We would like to thank Reviewer #3 for useful remarks and suggestions. We have done a further revision of the manuscript after round #2 . We believe that we have now addressed last concerns, improved the quality of our manuscript and hope that it is now suitable for publication in Sensors.
